# Distance Learning during the 2020 COVID-19 Lockdown: The Experience of Italian Middle School Students

**Lynda S. Lattke** , **Aurelia De Lorenzo \*** , **Beatrice Tesauri and Emanuela Rabaglietti**

Department of Psychology, University of Turin, via Verdi 10, 10124 Turin, Italy
* Correspondence: aurelia.delorenzo@unito.it

**Abstract:** Northern Italy was one of the first European regions to be affected by COVID-19 restrictions which led to school closures and the compulsion to learn from home. This article examines middle school students' experiences with distance learning to determine what they found most difficult, what they liked most and what they liked least during the 2020 lockdown. A total of 285 students (56% female; 44% male) with mean age of 13 years (±1 year; min = 11; max = 15) completed the online questionnaire. Responses to three open-ended questions were analyzed and coded using content analysis and an inductive approach. SPSS 26 was then used for descriptive analysis based on the frequencies of the categories that emerged: Learning, Device, Relationship, Other, Environment, Nothing, and Time. The results suggest that important aspects of students' lives during the lockdown had dual meanings. For example, technological devices were experienced as a means of communication, learning, and maintaining relationships, but were also associated with inequities, technical difficulties, and misunderstandings. Student responses support schools' role as a place to foster technological skills, especially social and emotional skills, in order to develop concrete strategies to assist students and teachers improve their relationship skills and be better prepared for future pandemics.

**Keywords:** middle school students; teachers; social–emotional skills; distance learning; content analysis

## 1. Introduction

The consequences of the COVID-19 health emergency have resulted in an unexpected life-changing event which has brought stress for most people at various levels. The strict measurements to contain the virus have altered the continuity in the lives of adults, teens, and children in entire communities [1]. One of these measures was the Italian government's order to close all schools and enact distance learning from home [2], a change which caught most teachers and students psychologically and technologically unprepared [3–6].

During this pandemic, school communities suddenly found themselves confronted with a completely new and unexpected scenario, moving from the traditional form of face-to-face teaching, which was essentially based on paper books and notebooks and face-to-face interactions and human contact, to an exclusive form of teaching/learning based exclusively on the use of digital platforms [7]. This new reality led to a number of problems: in addition to uncertainty about access to the necessary technological devices and programs for online instruction [6,8], a school day had to be reinvented in order to strengthen bonds within a "community" of staff, teachers, students, and parents who had to communicate through a computer or cell phone screen while trying to find a substitute for the informal style and warmth that is more readily available in person [9].

### 1.1. Schools' Role in the Development of Students' Social–Emotional Skills

School is a place where students spend most of their day, and, therefore, one of the social–relational settings where young people begin to construct their own personal identities [10], including the development and reinforcement of social–emotional skills [11,12]. In

fact, social–emotional skills, like general academic skills, continue to develop over time, but their promotion is particularly important during the preadolescent (11–13 years) and adolescent (14–18 years) periods [13]. These sensitive life stages are characterized by changes, opportunities, and important challenges that traditionally occur when physically present at school.

As a result, adolescence is a moment of reorganization of cognitive, emotional, and social systems [14,15], in which the influence of each component of the school community can be meaningful, in particular, the relationships with classmates and teachers. Teachers' role can be critical in motivating students to learn [16]. As Frymier and Houser [17] reported, it is not enough for teachers to be knowledgeable; it is equally important that they know how to communicate verbally and non-verbally [18]. Similarly, the relationship students build with their peers can influence their motivation to learn and encourage them to achieve their objectives [19–21]. All of these paradigms were challenged when schools were closed due to the COVID-19 pandemic.

*1.2. Consequences of Distance Learning on Students' Lives during the Lockdown*

Distance learning meant a reduction and/or elimination of contact with the senses, including physical contact, perception, looks, motor skills, and even smells, which drastically limited the active student–student/student–teacher relationship in the learning experience and may also have significantly affected cognitive development [22]. Considering the Italian context, where schools were neither technologically nor mentally prepared for this change from face-to-face to online instruction [5], the sudden closure of schools meant a steep learning curve for everyone involved [8,23]. Both teachers and students had to struggle with communicating through an electronic device and learn to deal with problems that ranged from interruptions in the internet connection to non-functioning microphones or cameras [24], situations that led to misunderstandings and, potentially, to subsequent demotivation to study [25]. Indeed, studies from the lockdown period confirm how much learning was affected by these changes and how difficult it was to adapt to a new way of teaching and learning [26]. This problem was exacerbated by the fact that middle school students were at home, an environment they associated with a place to do homework but not with school itself. In these circumstances where learning was at stake, teachers also felt pressured to complete the school curriculum at a certain pace, which caused stress and may have affected the relationship with their students [27]. In fact, during the COVID-19 pandemic, it has become quite evident the role schools play relative to students' non-academic needs, and the influence that these needs can have on learning [28].

On this basis, this qualitative study describes how distance learning was experienced by middle school students by collecting and analyzing their responses to three open-ended questions:

1. What do you find most difficult in distance learning?
2. What do you like most about distance learning?
3. What do you like least about distance learning?

## 2. Materials and Methods

*2.1. Participants*

A total of 285 middle school students from northwestern Italy (Piedmont) participated in this study. Students' mean age was of 13 years ($\pm$ 1 year; min = 11; max = 15) and of this total, 161 were female (56%) and 124 were male (44%). There were 79 students (28%) from the first year, 104 (36%) from the second year, and 102 (36%) from the third year of middle school.

*2.2. Procedure and Data Collection*

After our study was approved by the University Bioethics Committee (Prot. No. 157942), we distributed an online questionnaire in May 2020 asking students to consent to participate. Because all students in our sample were minors, parental/guardian consent was

also obtained. The questionnaire was distributed by school principals, who were asked to share the link to the questionnaire with their teachers. For distribution, we also relied on the Piedmont Network of Health promoting schools which collaborates with the European network (SHE-Schools for Health in Europe Network Foundation). The questionnaire included sections on sociodemographic data, daily routines, and eight ad hoc open-ended questions. Overall, it took 20′ to complete.

In this study, we examine students' responses related to distance learning and how it has affected their lives.

### 2.3. Data Analysis

Content Analysis: Determining the Codes and Categories

First, three inter-coders read the responses to each of the questions and independently decided on the codes based on the frequency of words and arguments [29]. The inter-coders then discussed and agreed on the categories.

The formulation of the categories was created while trying to respect as much as possible the criteria for mutual exclusivity and exhaustiveness; however, it was not possible to do so in several cases because specific responses reported content that applied to more than one category.

For this purpose, a coding grid was created to which all coders referred. Descriptive analysis of the coded data was carried out using SPSS.

Value labels in SPSS were determined based on whether the argument was either present (1) or absent (0). For example, responses in the Relationship category were assigned a value of 1 regardless of whether the response was positive: "*due to the lack of contact with teachers and classmates*" or negative: "*low availability from teachers*"; in both cases, the Relationship category was present. The following table shows a summary of the codes which led to the creation of each of the categories (Table 1).

**Table 1.** Summary of the coding grid for each of the categories.

| Categories | Description |
|---|---|
| **Device** | |
| Use of computer (hardware and software) | Understanding how to use the device and fix any technical problems. |
| Access to computer (hardware and software) | Possibility to follow lessons and stay in touch with classmates and teachers. |
| Quality of internet Connection | When poor, it led to misunderstanding amongst those using it. |
| Distraction | Due to internet connection, home environment or/and time online. |
| Health | Was affected due to the amount of time spent online. |
| **Learning** | |
| Technology | To follow online lessons and learn a new way of using electronic devices. |
| Understanding | Of material taught depending on a number of factors (i.e., online interactions, quality of connection). |
| Concentration level | Decreased due to distractions, yet it increased for those with a quieter home environment. |
| Anxiety | About exams if difficulty in understanding and keeping up with online lessons. |
| **Relationship** | |
| Use of computer screen | Developed a new perception of classmates and teachers due to being in different places. |
| Interactions | Changed dramatically due to distance learning and not having physical contact. |
| Communication | Was often perceived as trying to convey the subject being taught while the social dimension was reduced. |

**Table 1.** *Cont.*

| Categories | Description |
|---|---|
| Teachers' availability | In which the sense of immediacy and boundaries from their teachers changed. |
| **Environment** | |
| Comfort | As a result of following lessons from home and having a different sleep schedule. |
| Restrictive | Since schooling took place in a space they were not used to. |
| **Time** | |
| Management | Of time given that lessons were followed from home and there was more time to do other things. |
| **Other** | |
| Miscellaneous | Answers that were varied and did not fall in any of the categories. |
| **Nothing** | |
| No difficulties | Were found by these students. |

## 3. Results

### 3.1. Content Analysis

The following results are presented as individual categories based on responses to each question about what students found "most difficult," what they liked "most," and what they liked "least" about distance learning. At the end of each category, a student's response is reported for better illustration.

The content analysis resulted in the categories shown in Table 2:

**Table 2.** Categories based on responses to each question.

| | Categories | | | | | | |
|---|---|---|---|---|---|---|---|
| | **Learning** | **Device** | **Relationship** | **Environment** | **Time** | **Other** | **Nothing** |
| **Questions** | | | | | | | |
| 1. Difficulties | 142 | 90 | 48 | – | – | – | 35 |
| 2. Positive Aspects | – | 82 | 39 | 60 | – | 49 | 57 |
| 3. Negative Aspects | 70 | 116 | 90 | 17 | 24 | 67 | – |

Note: These cells indicate the absolute frequency (*n*) of the answers given inside each of the categories. Instead, the cells in which no numbers appear (–) indicate that these categories were not identified in the answers to each question.

### 3.1.1. What Do You Find Most Difficult in Distance Learning?

A total of four categories emerged for this question: Learning, Device, Relationships, and Nothing. Below is the description of each category:

- "Most Difficult" in Distance Learning: Learning (*n* = 142)

For 50% of students' responses, difficulties emerged particularly when following lessons at a distance. The reasons for this were varied, ranging from the difficulty of staying awake after so many hours of online classes in front of the computer screen, to distractions that occurred not only at home but often due to a poor internet connection that interrupted the flow of the lesson. Students also mentioned increasing difficulty in certain subjects such as mathematics and foreign languages. Students felt that it was difficult to discuss certain arguments further, which resulted in widening the knowledge gap that some of them already had. As a result, some students expressed anxiety about their written and oral exams and felt insecure about their overall learning. Example: "That sometimes you don't understand but you don't say it".

- "Most Difficult" in Distance Learning: Device (*n* = 90)

In total, 32% of students reported that they had technical difficulties using a device and felt unsure how to navigate the internet, such as how to find assigned homework or make sure the assignment had been properly sent to the teacher. Some participants reported having difficulties with the device, either because they had to follow the lessons with the device or because they had technical problems with certain parts such as the video or/and the microphone. Some of the difficulties focused on concentration interference due to the number of hours spent in front of a screen. Example: "Not everyone has a good internet connection, and it is difficult to concentrate".

For both categories, Learning and Device, 20 responses expressed difficulty in understanding the teacher's explanations, either because of the device, which was not only technically distracting, but also because of the added difficulty that students were not accustomed to learning through a computer screen (especially at the beginning of the closure, when distance learning had just begun). These difficulties were also experienced by those who had to watch the video lessons (i.e., not the live lessons). This response describes both scenarios: "*When your internet goes down and you can't finish the lesson. When it's windy outside and the internet is weak, it's difficult to understand what the teachers are saying since their voices are interrupted*".

- "Most Difficult" in Distance Learning: Relationship (*n* = 48)

Only 17% of students indicated that it was difficult not to see each other. In this category, 43 students specified these difficulties based on whether it was the teacher and/or the classmate:

-Relationship with teachers and students (*n* = 28): Overall, students indicated that it was difficult to physically see either their teachers or their classmates which resulted in the lack of interaction and socialization: "*The fact that we cannot see the teachers nor classmates in person so the atmosphere changes*".

-Relationship with classmates (*n* = 5): These responses were similar when they mentioned only their classmates: "*Not being able to hang out with my friends*".

-Relationship with teachers (*n* = 10): Students' responses focused on how communication with their teachers had changed; some felt that they were less available and that it took longer to receive answers from them. One student described it this way: "*. . . if you have a doubt you have to send a message and wait for the teachers to answer but when we were in class they could give you an answer right there and then*". Thus, the device also determined the communication dynamic: at a distance, students often had to wait for the teacher's response, whereas at school, the response was much more immediate. Others felt that a faulty internet connection made teachers suspicious, wondering whether or not the student was actually having problems with the internet. Being in different locations meant no longer having the opportunity to discuss things as they did in school, and many felt that communication was limited to the classroom: the social dimension had drastically decreased.

- "Most Difficult" in Distance Learning: Nothing (*n* = 35)

Finally, 12% of the responses included answers from students who had no difficulties during distance learning. Example: "*Quite frankly, it has not been difficult for me*".

3.1.2. What Do You Like Most about Distance Learning?

We identified a total of five categories: Device, Environment, Relationships, Other, and Nothing.

- "Liked Most" in Distance Learning: Device (*n* = 82)

For 29% of students, the benefits of the device were highlighted as a way to continue their education. For these students (*n* = 14), it was important and fun for them to learn how to use this type of technology and this new way of learning. Students also appreciated the use of technology for video calls and the opportunity to improve their technological skills by using computers and cell phones differently, including computer programs (such as

Word and PowerPoint). Example: "*The thing I like most about distance learning is to know how to use technology*".

- "Liked most" in Distance Learning: Environment (*n* = 60)

In this category, 21% of the responses described the environment as the convenience of being able to stay at home in a quieter place and the convenience of being able to get up later because of not having to physically go to school. Some students mentioned that they appreciated that there was less chaos because they were in a familiar environment. This allowed them to enjoy the silence, which improved their concentration and made them feel more comfortable during exams: "*That I'm home and I don't have to take the bus and I don't have to get up early*".

- "Liked Most" in Distance Learning: Relationship (*n* = 39)

A total of 14% of students' responses referred to the relationship. A more detailed analysis of this category indicates the following breakdown:

-Relationship with teachers and classmates (*n* = 34): Most students were grateful to be able to see their classmates and teachers, even if it was only through a computer screen: "*The fact that I have never been so eager or thrilled to see my classmates or professors again after so long of not seeing them even if through a screen*". A few appreciated the opposite and were content not to see some of their classmates: "*Staying home, and not seeing some of my classmates every day*".

-Relationship with teachers (*n* = 5): Some students felt that teachers were more available than usual, i.e., outside of class time, and that they could talk to them about anything. These students appreciated the flexibility of their teachers: "*That there is plenty of time to do homework and l can contact teachers 24/7 (if they respond)*".

A total of 32 students were grateful to have access to a device that allowed them to see their classmates and teachers, as mentioned earlier, even if it was a relationship mediated by the device.

- "Liked Most" in Distance Learning: Other (*n* = 49)

A total of 17% of responses ranged from having more time to look up information to being more independent and organized to being able to cheat more easily and enjoy the reduction in homework. Some students (*n* = 13) were also pleased with the shorter length of lesson time.

- "Liked Most" in Distance Learning: Nothing (*n* = 57)

Finally, 20% of these students thought that there was nothing positive about the distance-learning experience: "*Nothing, I don't like it at all*".

3.1.3. What Do You Like Least about Distance Learning?

The categories that emerged were: Device, Relationship, Learning, Time, Environment, and Other.

- "Liked Least" in Distance Learning: Device (*n* = 116)

The word "connection" predominated in this category with 41% of responses. Students described how often they had problems with the quality of the connection and, thus, with the number of interruptions. Responses also mentioned technical problems, ranging from whistling noises due to open microphones to problems with the video camera. In addition, teachers in a particular class were using different platforms, which was confusing for students when they needed to find or send homework. Some of the responses were about the impact of the device on health, as sitting in front of the screen for long periods of time caused headaches and eye irritation: "*Sometimes maybe the connection doesn't work well and I can't understand what the teachers are saying and…as a result, I cannot study well*".

- "Liked Least" in Distance Learning: Relationship (*n* = 90)

In 32% of the responses, the relationship was somehow determined by the device. They mentioned the coldness they felt from the computer screen, since there was no physical contact, and that sometimes they could not see all their classmates in every lesson, since not all of them could follow the online lessons. A more detailed analysis of this category indicates the following breakdown:

-Relationship with classmates (*n* = 80): Overall, students indicated that they miss the feeling of being a class group and not having their friends next to them to talk to: "*I do not see them in real life*". One of the aspects students liked least was not being able to see their classmates in person: "*I cannot see my friends and the internet connection may drop*".

-Relationship with teachers (*n* = 10): These students' responses were about the difficulty of not seeing their teachers and not interacting, discussing, and communicating with them as they used to when they were present in school: "*Lack of direct contact with the teachers*".

- "Liked Least" in Distance Learning: Learning (*n* = 70)

About 25% of the students expressed concern about how they would be able to cope with the exams because the level of attention and comprehension in distance learning was different from that in school. One student explained this as follows: "*. . . sometimes I don't understand the lectures and since they are too short I don't have time to ask* (teachers) *to explain* (the concepts) *again*". As in the analysis of what was most difficult in distance learning in terms of the Learning category (see Section 1.2.), the answers were similar: even when there were explanations, they could not always follow them because of the poor internet connection. The word "understanding" was repeated frequently: students were concerned about how little they could understand the online lessons compared to the lessons in school.

- "Liked Least" in Distance Learning: Time (*n* = 24)

Only 8% of students complained about less lesson time and about the distribution of lessons throughout the day compared to before COVID-19. These students also complained that teachers had less time to explain the material accurately because lessons were shorter; they also reported having more homework. This meant that students had to organize and allocate their time differently than they had before the pandemic: "*I have difficulty managing my time*".

- "Liked Least" in Distance Learning: Environment (*n* = 17)

In contrast to the results under "Most Liked" in distance learning (2.2), 6% of respondents did not appreciate following lessons from home and felt a lack of freedom and of daily routine: "*. . . I cannot touch my desk and wander around in the classroom (as I used to)*".

- "Liked Least" in Distance Learning: Other (*n* = 67)

Finally, 24% of students' responses did not fall in any of the above categories.

## 4. Discussion

The results of our study reflect the impressions of middle school students in Northern Italy regarding distance learning during the first closure in spring 2020. According to content analysis, "Learning," "Device," and "Relationships" were the most frequent categories that emerged from all three open-ended questions answered by students, while "Environment" and "Time" were less frequent.

The Device category is the most prevalent in the responses students provided to all three questions. The tool that made distance learning possible seems to represent a topic on which students show an ambivalent attitude. On the one hand, students identify it as one of the main difficulties they encountered during the first lockdown (32% of responses), and as one of the aspects they liked least about distance learning (41% of responses). On the other hand, however, some students (29% of the responses) stated that the device represented one of the most interesting, innovative, and enjoyable aspects of distance learning.

This ambivalence about the device can be explained by a number of factors concerning Italian students and schools in the pre-COVID period. As the literature suggests, Italy, unlike other European countries, was not prepared for the sudden technological immersion, mainly because schools did not use the device regularly before the lockdown [5]. Although the students who participated in the study belong to the generation known as digital natives, that is, they were born and raised with digital tools to access the internet [30], several studies have shown that being a digital native does not necessarily imply possessing good digital skills [31,32]. In fact, there are many disparities between the digital skills of students in this age group. Moreover, it should be considered that many of their digital skills are related to recreational use such as social networking and online gaming [33]. For these reasons, we think that the results in the Device category are interesting: students enjoy using these types of devices, but they need adequate training to use them competently in the school context.

The Learning category, which appears only in the responses to the questions about the difficulties (in 50% of the responses) and what they liked least (in 25% of the responses) related to distance learning, is closely linked to the device. Indeed, most of the learning problems reported by students can be attributed to difficulties in using or a lack of availability of suitable devices. As mentioned above, distance learning based on new technologies was not introduced in Italian schools before the pandemic period. This meant that neither teachers nor students were prepared for this situation, which made the transition to this new way of learning much more difficult.

The Digitization of Economy and Society Index (DESI) confirms the serious gap in digital skills in Italy, which ranks 20th in Europe [34]. The fact that students did not have easy access to digital devices and did not know how to use them made the situation worse, as students had to spend many hours in front of screens [24] while trying to figure out how to use a device and follow a lesson at the same time. As a result, the learning mediated by the device was more complex, as students reported in their responses.

The Relationship category is the third most important and appears as a response to all three open-ended questions: as "difficulty" (for 17% of responses), as "liked most" (for 14% of responses), and as "liked least" in distance learning (for 32% of the responses). This category includes, in particular, responses related to the relationship with classmates and teachers that students were able to maintain thanks to distance learning. However, the presence of this category in all three questions leads to reflections on the role of distance learning.

Relationships, as students were accustomed to living them before the pandemic, were undoubtedly the most severely affected during the lockdown period. The inability to socialize with peers and teachers was one of the aspects that students most often report as a reason for their great distress. Several studies indicate that the lack of living relationships in physical presence has played a role in exacerbating anxiety disorders, stress, and risk behaviors in students [35–37]. It is not surprising, then, that the relationships mediated by the device were perceived as complex, a situation that led to misunderstanding, difficulty, and isolation.

At the same time, during the period in which the students answered our questionnaire, for more than three months, the device was the only instrument that could guarantee students a daily relationship with people outside their family. For this reason, the relationship is one of the "most liked" aspects of distance learning, considering that in the preadolescent and adolescent age, socialization in school is fundamental for the development of one's identity [10].

Even if less represented in the responses, it is interesting to note that for both categories of Environment and Time, a new and double meaning was acquired during the lockdown period [38,39]. Again, there was an ambivalent presence for both categories. For some of the students, the opportunity to learn from home, surrounded by their own comforts, was a positive aspect. Given the shorter lesson time and delayed school start, they were able to sleep more and spend more time with their families and in extracurricular activities when

they would have otherwise been at school. These were all factors that meant that the home environment was particularly valued, but at the same time brought with it many more distractions that interfered with learning, including having to share reduced home space with other family members. Although the latter did not feature strongly in these responses, studies such as Parolin and Lee [40] report the exacerbation of inequities among students who had different resources (in terms of equipment) during the period of initial lockdown, as well as spaces that were not conducive to learning (i.e., did not have a dedicated room), were often noisy and not very bright, and had to be shared.

Although our study reveals important aspects of middle school students' perceptions and experiences of distance learning in Northwestern Italy during the lockdown period, it has some limitations. One of them is that we did not use qualitative data analysis software that may have led to further understanding of students' responses. We are also aware that our study only reflects the students' views. It would have been interesting to collect teachers' views on the same questions to understand how similar or different these two key components of the school community experienced online learning/teaching during this time.

## 5. Conclusions

During the 2020 school closure, students faced many unexpected challenges that redefined their interactions with teachers and classmates. In our study's findings, students distinguish between what they liked most and what they liked least or found most difficult about distance learning.

They tell us how their learning experience was affected by the novelty of using electronic devices, which took on a new meaning: from an entertainment tool to a learning medium. Despite the ambivalence associated with the use of digital tools, students report the excitement associated for many with the opportunity to acquire technological skills needed in the world in which we live. However, what inequities lie behind the inability to guarantee all students the same digital tools? The learning of digital skills and the normal continuation of educational and training activities are still tied to the possession and availability of tools that are not always accessible to all students. Not all schools are able to support families who do not have access to digital tools, further deepening the social divide.

During this time, important aspects of life such as time, home, and relationships at school, as well as relationships with the school itself, also took on new meaning. Most students recognized how important it was for them to physically go to school and interact with teachers and classmates. Their responses and comments indicate that while they appreciated the extra time they could spend at home, they were also concerned about the consequences of fewer hours of lesson time and homework. However, it was a complex period that gave students, even the youngest, a greater awareness of themselves and their role as students, as well as an understanding and appreciation of the intrinsic educational value of the school environment. Students soon realized that following lessons from home is not the same as learning at school. The context and relationships associated with this type of learning were compromised by social distancing.

The results of the present study illustrate how communication via a device can impact not only student learning, both positively and negatively, but also perceptions of relationships. In students' perceptions, all of the difficulties they faced during the lockdown period emerged, but so did the positive aspects, such as the value of relationships, both with peers and adult caregivers, including the growing interest in skills related to the use of new technologies. For these reasons, it is desirable to continue to actively support school-based education that promotes technological skills among teachers and students. At the same time, it is important to foster social–emotional skills in young people so that they can develop better communication skills and understanding.

The findings of our study are a call to action for schools to actively seek strategies that promote the relational health and well-being of students and teachers, especially in challenging times such as those we are currently experiencing.

**Author Contributions:** Conceptualization, L.S.L., A.D.L. and E.R.; methodology, L.S.L., A.D.L. and E.R.; writing—original draft preparation, L.S.L., A.D.L., B.T. and E.R.; writing—review and editing, L.S.L., A.D.L., B.T. and E.R.; supervision, E.R. All authors have read and agreed to the published version of the manuscript.

**Funding:** This research received no external funding.

**Institutional Review Board Statement:** The study was conducted in accordance with the Declaration of Helsinki, and approved by the Ethics Committee of The University of Turin (protocol N. 157942 on 15 April 2020).

**Informed Consent Statement:** Informed consent was obtained from all subjects involved in the study.

**Data Availability Statement:** The data presented in this study are available upon request from the SE-CREA research group at the University of Turin. Data access requests can be made by contacting Emanuela Rabaglietti at emanuela.rabaglietti@unito.it.

**Acknowledgments:** We would like to thank the students who participated in this study and their teachers and the school directors for their support in facilitating the data collection. We would also like to thank Marisol Zanfabro for her contribution in the early stages of our research.

**Conflicts of Interest:** The authors declare no conflict of interest.

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
