# Peer review of "Distance Learning during the 2020 COVID-19 Lockdown: The Experience of Italian Middle School Students"

_adolescents, doi:10.3390/adolescents2030030_

Round 1

Reviewer 1 Report

The conception, conduct and description of the research are excellent. I have just one suggestion to improve the readability of the Discussion and Conclusions sections by dividing them into shorter paragraphs. The following list contains my suggestions for the beginning line of each paragraph:

Discussion

The results of our study …

The Device category is …

This ambivalence about the …

The Learning category, which …

The Digitization of Society and Economy Index …

The Relationship category is …

Relationships, as students were accustomed to …

At the same time, during …

Although less represented in the responses …

Although our study reveals …

Conclusions

During the 2020 lockdown …

During this time, important aspects …

The results of this study …

The conclusions of our study … 

Author Response

Dear Reviewer,

Thank you for your positive feedback, we really appreciate it. We also thank you for your suggestions which we have included in the article. There are now several changes which had been suggested by two other reviewers. With all of your suggestions, we believe the article has improved.

Reviewer 2 Report

Please check the below suggestions:

- In the abstract, the sample is mentioned making only reference to the female gender and not to the male gender there is an omission of the latter.

- The abstract needs to be reformulated since it is unclear and gives a vague idea about the study.

- Review the citations according to the standards of the journal. [3. 4; 5, 6]. In the text, reference numbers should be placed in square brackets [ ], and placed before the punctuation; for example [1], [1–3] or [1,3]. For embedded citations in the text with pagination, use both parentheses and brackets to indicate the reference number and page numbers; for example [5] (p. 10). or [6] (pp. 101–105).

- Review punctuation of the entire document

- Avoid redundancy in the same paragraph on page 2 (Indeed) and on page 3

- In the section Procedure and Data collection, the application time of the questionnaire is not mentioned

- Check if the definition of the initials is correct: the SHE network (Piedmont Network of Health promoting schools)

- It would be appropriate to include: a coding grid to be an important support for data analysis.

- It is suggested to have used a purely qualitative analysis with a tool of this type instead of SPSS, despite having mentioned it in the limitations. - Check the Title of the table

- Check wording, grammar, there are unclear paragraphs that lead to confusion of information. Page 5 (check space, paragraph alignment)

- Pg 6 (see 1.2.), it is recommended not to mention this epigraph number, but rather a concise summary of it.

- Avoid redundancies page 7: For this reason, the relationship is one of the most "most liked"

- Avoid redundancy page 8: of our study, students distinguish

- The conclusions do not include citations:[39]. [5]. In the conclusions there is no need to add data or bibliographic citations, the only thing that goes is the condensed summary of what we have done throughout the work.

- Check the references according to the format suggested by the journal. References should be described as follows, depending on the type of work:

·        JournalArticles:
1. Author 1, A.B.; Author 2, C.D. Title of the article. 
Abbreviated Journal Name YearVolume, page range.

·        Books and Book Chapters:
2. Author 1, A.; Author 2, B. Book Title, 3rd ed.; Publisher: Publisher Location, Country, Year; pp. 154–196.

- The format does not allow for an adequate suggestion of change as each line is not numbered.

Author Response

Dear reviewer, 

Thank you for your comments, we believe the article has improved.

Please see the attached file with the revised manuscript.

Q1- In the abstract, the sample is mentioned making only reference to the female gender and not to the male gender there is an omission of the latter.

A1-Thank you for your suggestion. However, due to a word count limit of 200 words, we only highlighted the percentage of the most prevalent gender

Q2- The abstract needs to be reformulated since it is unclear and gives a vague idea about the study.

A2- Thank you for your suggestion. We hope this abstract is now clearer. In the manuscript you will see the old and the new version of the abstract.

Q3- Review the citations according to the standards of the journal. [3. 4; 5, 6]. In the text, reference numbers should be placed in square brackets [ ], and placed before the punctuation; for example [1], [1–3] or [1,3]. For embedded citations in the text with pagination, use both parentheses and brackets to indicate the reference number and page numbers; for example [5] (p. 10). or [6] (pp. 101–105).

A3-Thank you, we have reviewed all citations to make sure they follow the journal’s requirements.

Q4- Review punctuation of the entire document.

A4-Before we submitted the article to Adolescents, the punctuation of the document had been reviewed. Based on the positive feedback of one of the reviewers, we kindly ask you to provide us with a couple of examples where you would suggest a change. Thank you

Q5- Avoid redundancy in the same paragraph on page 2 (Indeed) and on page 3

A5-Thank you, we have substituted the second ‘indeed’ with ‘in fact’.

Q6- In the section Procedure and Data collection, the application time of the questionnaire is not mentioned

A6-In this section we had already mentioned that the data collection took place in May 2020. Thanks to your suggestion, we have added the time it took to complete the questionnaire

Q7- Check if the definition of the initials is correct: the SHE network (Piedmont Network of Health promoting schools)

A7-We have clarified the acronym.

Q8- It would be appropriate to include: a coding grid to be an important support for data analysis.

A8-We have now included a table in which we summarize the coding grid. Furthermore, we have added a new paragraph in the Materials and Methods/Data analysis section in which we clarify the use of SPSS

Q9- It is suggested to have used a purely qualitative analysis with a tool of this type instead of SPSS, despite having mentioned it in the limitations.

A9-Thank you for your suggestion. In fact we do highlight not having used a software for qualitative analysis as a limitation in our work since we are aware that it can assist in the process and we will certainly use it on another occasion.

Q10- Check the Title of the table

A10-The title is now correctly placed

Q11- Check wording, grammar, there are unclear paragraphs that lead to confusion of information. Page 5 (check space, paragraph alignment)

A11-We have reviewed the spacing and alignment of the various paragraphs of the article, thank you.

Q12- Pg 6 (see 1.2.), it is recommended not to mention this epigraph number, but rather a concise summary of it.

A12- We have modified it so that it now reads: (see section 1.2. Consequences of Distance Learning on Students’ Lives During the Lockdown)

Q13- Avoid redundancies page 7: For this reason, the relationship is one of the most "most liked"

A13- We eliminated the first ‘most’

Q14- Avoid redundancy page 8: of our study, students distinguish

A14-We changed it to: “In this study's results, students…” so that ‘study’ and ‘students’ are not next to each other.

Q15- The conclusions do not include citations:[39]. [5]. In the conclusions there is no need to add data or bibliographic citations, the only thing that goes is the condensed summary of what we have done throughout the work.

A15-We have reviewed the conclusions and changed them. Since there are several changes, it is all highlighted in yellow. We believe this version summarizes better the essence of the article.

Q16- Check the references according to the format suggested by the journal. References should be described as follows, depending on the type of work:

  • JournalArticles:
    1. Author 1, A.B.; Author 2, C.D. Title of the article. Abbreviated Journal NameYearVolume, page range.
  • Books and Book Chapters:
    2. Author 1, A.; Author 2, B. Book Title, 3rd ed.; Publisher: Publisher Location, Country, Year; pp. 154–196.

A16- The references are now written according to the journal’s requirements

Q17- The format does not allow for an adequate suggestion of change as each line is not numbered.

A17-We used the template provided by the journal which was not numbered.

Reviewer 3 Report

The scientific article entitled "Distance education during confinement by Covid-19 of 2020: the experience of Italian high school students" brings together a number of interesting aspects related to the open access journal on adolescents, it is also a relevant topic and contributes to improve teaching in times of pandemic.

However, it is suggested to add new relevant references on the object of study related to the consequences of Distance Education on students' lives during confinement. In this sense, it is advisable to consult the Web of Science and SCOPUS databases.

On the other hand, the use and application of statistical software for the analysis of semantic networks of qualitative data, such as Nivo, Aquad, Atlas.ti, etc., is recommended.

Finally, the conclusions require a more in-depth approach and it is recommended that the usefulness and contribution of the study to the scientific community be explored in greater depth.

Author Response

Dear Reviewer,

Please find below answers to each of your comments:

Q1- However, it is suggested to add new relevant references on the object of study related to the consequences of Distance Education on students' lives during confinement. In this sense, it is advisable to consult the Web of Science and SCOPUS databases.
A1- Thank you for your suggestion, we have added more references and highlighted them
Lee, J.; Lim, H.; Allen, J.; Choi, G. Effects of Learning Attitudes and COVID-19 Risk Perception on Poor Academic Performance among Middle School Students. Sustainability 2021, 13, 5541. doi: 10.3390/su13105541
Sofianidis, A.; Meletiou-Mavrotheris, M.; Konstantinou, P.; Stylianidou, N.; Katzis, K. Let Students Talk about Emergency Remote Teaching Experience: Secondary Students’ Perceptions on Their Experience during the COVID-19 Pandemic. Educ. Sci. 2021, 11, 268. doi: 10.3390/educsci11060268

Q2- On the other hand, the use and application of statistical software for the analysis of semantic networks of qualitative data, such as Nivo, Aquad, Atlas.ti, etc., is recommended.

A2- Thank you for your suggestion. In fact we do highlight not having used a software for qualitative analysis as a limitation in our work since we are aware that it can assist in the process and we will certainly use it on another occasion.

Q3-Finally, the conclusions require a more in-depth approach and it is recommended that the usefulness and contribution of the study to the scientific community be explored in greater depth.
A3-We have reviewed the conclusions and changed them. Since there are several changes, it is all highlighted in yellow. We believe this version summarizes better the essence of the article.

Round 2

Reviewer 2 Report

The below suggestions are not taken into consideration please check and proceed accordingly:

Q1- In the abstract, the sample is mentioned making only reference to the female gender and not to the male gender there is an omission of the latter.

A1-Thank you for your suggestion. However, due to a word count limit of 200 words, we only highlighted the percentage of the most prevalent gender

  • Despite the limitation of words gender is an important piece of information.

Q3- Review the citations according to the standards of the journal. [3. 4; 5, 6]. In the text, reference numbers should be placed in square brackets [ ] and placed before the punctuation; for example, [1], [1–3] or [1,3]. For embedded citations in the text with pagination, use both parentheses and brackets to indicate the reference number and page numbers; for example, [5] (p. 10). or [6] (pp. 101–105). Pag 2 [23, 8]. [8,23]

2. Materials and Methods

2.1. Participants

The male gender is not mentioned. How many male students have participated in the study???

Q15- The conclusions do not include citations:[39]. [5]. In the conclusions, there is no need to add data or bibliographic citations, the only thing that goes is the condensed summary of what we have done throughout the work.
